# Exploring the Effect of Preoperative Stereopsis on Visual Outcomes in Hyperopic Presbyopes Treated with PresbyOND^®^ Laser Blended Vision Micro-Monovision

**DOI:** 10.3390/jcm12196399

**Published:** 2023-10-07

**Authors:** Julia Hernández-Lucena, Federico Alonso-Aliste, Jonatan Amián-Cordero, José-María Sánchez-González

**Affiliations:** 1Department of Physics of Condensed Matter, Optics Area, University of Seville, Reina Mercedes S/N, 41012 Seville, Spain; julherluc@alum.us.es; 2Department of Ophthalmology (Tecnolaser Clinic Vision^®^), Refractive Surgery Center, Juan Antonio Cavestany, 41018 Seville, Spain

**Keywords:** PresbyOND^®^ Laser Blended Vision, Preoperative Stereopsis, hyperopic presbyopes, micro-monovision, refractive treatment efficacy, Uncorrected Distance Visual Acuity (UDVA), Corrected Distance Visual Acuity (CDVA), binocular summation

## Abstract

We investigated the effects of Laser Blended Vision (LBV) on binocular summation and stereopsis in the treatment of presbyopia and hyperopia. Using a unidirectional, retrospective longitudinal design, data from 318 patients who underwent the Zeiss PresbyOND^®^ LBV surgical procedure at Tecnolaser Clinic Vision Ophthalmology Center in Seville, Spain, were analyzed. The findings indicate that stereopsis quality significantly influenced short-term post-operative visual outcomes in measures like Uncorrected Distance and Near Visual Acuity (UDVA and UNVA). However, the impact of stereopsis on visual outcomes appeared to diminish over time, becoming statistically insignificant at the 12-month post-operative mark. The study suggests that while stereopsis is a crucial factor in the short term, its influence on visual outcomes tends to wane in the long-term postoperative period. Future studies are essential to elucidate the enduring clinical ramifications of these observations.

## 1. Introduction

In the continuously advancing sphere of ophthalmology, our research venture aims to illuminate an area that has thus far remained obscure: the intricate interplay between Preoperative Stereopsis and the effectiveness of PresbyOND^®^ Laser Blended Vision in treating those who are hyperopic Presbyopes. This study offers a nuanced, much-needed perspective on the emerging methodologies in presbyopia treatment, a field that is already replete with rapidly evolving technologies and approaches.

To clarify, stereopsis is a complex visual function that allows individuals to perceive the world in three dimensions, resulting from the brain’s ability to integrate different images from each eye into a single, depth-laden visual field [1]. Stated differently, stereopsis refers to the visual ability to perceive depth by integrating two slightly different images from each eye into a single, three-dimensional representation [2]. This faculty enriches the visual experience and is vital for executing a multitude of everyday activities. Hyperopic presbyopes are individuals who, in addition to suffering from presbyopia, have hyperopia, or farsightedness [3]. PresbyOND^®^ Laser Blended Vision is an advanced surgical technique that merges elements of traditional monovision with specially induced spherical aberrations, thereby achieving an extended range of focus [4].

Historically, the surgical domain has witnessed extensive research on presbyopia, resulting in a host of interventions ranging from PresbyLASIK to corneal inlays [3,5,6]. Other pioneering methodologies, such as the utilization of Multifocal IOLs and laser-focused techniques like PresbyMAX and Custom-Q, have shown great promise. However, one conspicuous gap in current knowledge is the influence of preoperative visual conditions—most prominently, stereopsis—on the outcomes after surgical intervention [7]. It is important to note that hyperopic presbyopes constitute a significant segment of the population grappling with presbyopia, yet the existing literature seldom pays focused attention to them [8].

The historical evolution of treatment solutions for hyperopic presbyopes reveals a winding but nonetheless forward-moving path. Initial remedies were largely temporary and cumbersome, represented by options such as reading glasses and monovision contact lenses, which were not only palliative but also posed challenges for long-term use [9]. The advent of laser surgical procedures like LASIK monovision provided a glimmer of hope for more enduring solutions [10]. However, even these advanced options had limitations, including a decrease in contrast sensitivity and stereopsis. The introduction of PresbyOND^®^ Laser Blended Vision was thus a watershed moment that involved the ingenious use of micro-monovision techniques combined with induced spherical aberration to improve depth of focus [4,11,12,13,14,15].

In recent years, the notion of binocular summation has become a focal point in ophthalmological research [1]. This phenomenon—the visual system’s ability to compile information from both eyes into a perception more effective than that rendered by a single eye—has crucial implications for the treatment of presbyopia, particularly when technologies like Laser Blended Vision are being employed [16].

Often overshadowed in discussions of presbyopia treatment is the concept of stereopsis, a crucial part of our visual system that enhances our depth perception. Stereopsis, in turn, plays a critical role in facilitating binocular summation. When these elements are compromised, as they often are in traditional monovision treatments, the patient’s overall quality of vision post-operation can be negatively impacted. Consequently, an in-depth exploration of these often-overlooked elements can be the key to developing more efficient surgical interventions [14].

In Laser Blended Vision treatments, the conventional micro-monovision strategy is employed: one eye is conditioned for far sight while the other is tailored for near vision [13]. Although this approach has been efficacious in improving visual acuity, our understanding of the role that preoperative and postoperative stereopsis plays in defining its success remains fragmented. Recent research, such as the work by Uji et al. [17] has expanded the understanding of stereopsis by investigating its manifestation in monocular situations. Their pilot study suggests that the phenomenological experience of ‘realness’ in stereopsis could be dissociable from depth perception based on binocular disparity. Preliminary data suggests that a keen focus on maintaining or even enhancing stereopsis could significantly elevate the efficacy of Laser Blended Vision [11,12,13,15]. In line with this, Cheng et al. [18] found that normal observers were able to judge equidistance with high accuracy and precision using static and dynamic monocular depth information, suggesting that monocular vision may also be effective for distance-dependent tasks in natural environments.

This study aims for dual objectives: an assessment of the immediate effects of Laser Blended Vision on the phenomenon of binocular summation, and a detailed evaluation of the role that stereopsis plays in visual acuity recovery following surgical intervention. The associated hypotheses posit that Laser Blended Vision will lead to substantial improvements in binocular summation and that the presence of stereopsis will positively correlate with improved visual acuity in the postoperative period. Through an exhaustive examination of these often-neglected yet vital elements, we endeavor to contribute new insights to the existing academic corpus, thereby providing practitioners with critical information to refine treatment methodologies for presbyopia.

## 2. Materials and Methods

### 2.1. Study Design

This study is a retrospective, longitudinal clinical investigation targeting presbyopic and hyperopic patients. It aims to rigorously evaluate the effectiveness of the treatment for this specific demographic.

### 2.2. Scope and Duration

This retrospective study encompasses patient data spanning from January 2016 to December 2021, offering a comprehensive longitudinal perspective on the outcomes of the surgical procedure. By capturing multiple years of clinical data, the study aims to identify both immediate post-operative outcomes as well as long-term trends in efficacy and safety, thereby enriching the current scientific literature on the subject.

### 2.3. Participant Briefing and Consent

Prior to the surgical procedure, all participants received exhaustive briefing sessions encompassing both oral presentations and written informational materials. These sessions were designed to provide an all-encompassing understanding of what the surgery entails, including but not limited to, potential risks, expected benefits, postoperative care, and complications. Post-briefing, written informed consent was collected from all participants to validate their voluntary participation legally and ethically in the study.

### 2.4. Ethical Adherence and Institutional Review

The study is conducted under the stringent ethical guidelines stipulated by the Declaration of Helsinki, which outlines ethical principles for medical researchers involved in human clinical trials and other research studies. Further strengthening the ethical foundations of the study is the approval received from the Institutional Review Board (IRB) of Virgen Macarena and Virgen del Rocio University Hospitals. This approval, denoted by the promotion code 0863-N-22, serves as a robust ethical endorsement of the study’s methodology, objectives, and clinical relevance.

### 2.5. Data Protection and Patient Confidentiality

In compliance with privacy regulations and ethical guidelines, all patient data utilized in the study are anonymized and securely stored. Access to the data is strictly restricted to authorized research team members, ensuring that patient confidentiality remains uncompromised throughout the duration of the study.

### 2.6. Participant Demographics and Clinical Setting

The study focuses on a cohort with presbyopia and hyperopia, including those with astigmatism. Surgeries took place at Tecnolaser Clinic Vision in Seville, Spain, using data collected from 2016 to 2021. Post-operative monitoring extended up to 12 months to assess short and long-term outcomes.

#### 2.6.1. Detailed Inclusion Criteria

Gender and Age Range: The study welcomed participants assigned male or female at birth who were aged between 40 and 60 years at the time of undergoing the PresbyOND^®^ LBV technique.Clinical Condition: Participants should have had a clinical diagnosis of presbyopia, which may or may not coexist with hyperopia and/or astigmatism, substantiated by complete optometric and ophthalmological examinations.Refraction Parameters: Subjective measurements were capped at 5.00 D for hyperopia and 6.00 D for astigmatism to focus on moderate cases.Visual Acuity: A minimum Corrected Distance Visual Acuity (CDVA) of 0.10 LogMAR in the oculomotor-dominant eye was a prerequisite for inclusion in the study.Stereopsis: Normative stereopsis levels were maintained below 400 arc seconds as verified via standardized stereoscopic vision tests.

#### 2.6.2. Comprehensive Exclusion Criteria

History of Refractive Surgeries: Individuals who have undergone any form of refractive surgery in the past were categorically excluded from the study to prevent confounding variables.Amblyopia: Individuals demonstrating a documented presence of amblyopia with a Visual Acuity (VA) score equal to or less than 0.20 LogMAR were considered ineligible.Other Ophthalmic Conditions: Patients with conditions such as glaucoma, macular degeneration, retinal detachment, corneal disease, or any other severe ophthalmic issues that could confound the study’s outcomes were excluded. Pregnancy and Lactation: Women who were pregnant or lactating at the time of the planned surgical intervention were not considered for inclusion, given the potential for hormonal variations to influence visual acuity and refractive errors.Systemic Illness: Participants with systemic illnesses like diabetes, autoimmune disorders, cardiovascular diseases, or any condition affecting wound healing and visual outcomes were not included.

The study’s cohort and eligibility criteria aim for a homogeneous sample to rigorously assess the PresbyOND^®^ LBV technique’s efficacy and safety across a specific patient demographic.

### 2.7. Pre-Operative Assessments

The pre-operative evaluations were structured as an exhaustive set of diagnostic tests, built upon both optometric and ophthalmological frameworks. This approach provides a comprehensive understanding of each participant’s visual condition, thereby allowing for a highly individualized treatment plan.

Optometric Evaluations: Advanced autorefractometers (KR8800, Topcon, Tokyo, Japan) were used to provide quick and accurate measurements of the refractive errors in both eyes.Ophthalmological Assessments: A combination of Pentacam AXL^®^ (Oculus, Wetzlar, Germany) and Optical Coherence Tomography (OCT) (OftalTech, Barcelona, Spain) were utilized to assess the anterior and posterior segments of the eye. This included an evaluation of corneal topography, lens transparency, and retinal integrity.Additional Evaluations: Slit lamp biomicroscopy (LH-2000, Indo, Barcelona, Spain) and fundoscopy post-cycloplegic dilation were performed to scrutinize the ocular health further and rule out any co-existing pathology.

### 2.8. Eye Dominance Protocols

Sensory Dominance: The sensory eye dominance was meticulously assessed using a blurriness index. This required the addition of a +1.00 lens to each eye’s current subjective refraction, which facilitated the differentiation in the eye contributing most to distance binocular vision [19].Motor Dominance: The hole-in-card method was employed for determining the motor dominance of the eye. This test identifies the eye that is primarily responsible for guiding hand and body movements [20].

### 2.9. Micro-Monovision Tolerance Evaluation

We utilized the TITMUS test (2007, Vision Assessment Corporation, Elk Grove Village, IL, USA) to assess stereopsis, providing a reliable measure of depth perception. The test employs polarized images and graded circle tests to quantify stereoscopic acuity in seconds of arc [21]. Before the surgery, each participant’s ability to adapt to micro-monovision was rigorously assessed. Adaptive test glasses and a series of different lens combinations were used to simulate possible post-operative scenarios. This allowed the research team to predict the range of outcomes and adjust treatment planning accordingly [22].

### 2.10. Surgical Procedures and Professional Expertise

The surgical interventions were conducted by seasoned, board-certified ophthalmological surgeons, who are experts in the field of presbyopia laser correction techniques. The surgeries utilized state-of-the-art VisuMax femtosecond and Mel 90 excimer lasers. Pre-operative planning was meticulously performed using the Custom Refractive Software (CRS) Master (Zeiss, Jena, Germany).

### 2.11. Advanced Statistical Framework

Statistical Tools: SPSS version 26.0 was utilized for the statistical analyses. Visual Acuity measurements were standardized to the Snellen format, enhancing data consistency. Statistical Tests: Parametric dependent variables were scrutinized using Student’s *t*-tests. All inferential statistics were framed within a 95% confidence interval, which provides a robust method for hypothesis testing and data interpretation.

## 3. Results

### 3.1. Demographic and Baseline Characteristics

A total of 636 eyes from 318 patients, with a mean age of 51.05 ± 4.71 years (ranging from 40.00 to 60.00 years), met the study’s inclusion and exclusion criteria. The sample consisted of 124 males (39.00%) and 194 females (61.00%). All patients completed a six-month follow-up, and 320 eyes (51.11%) underwent an annual follow-up. The distribution of motor dominance was skewed towards right-eye individuals (62.9%) compared to left-eye individuals (37.1%). Similarly, sensorial dominance favored the right eye in 61.6% of the cases, whereas the left eye was dominant in 37.4% of the subjects. Stereopsis across the cohort had a mean value of 79.92 ± 80.46 s arc, ranging from 20 to 400 s arc.

### 3.2. Visual and Refractive Outcomes

Detailed preoperative, six-month, and twelve-month visual acuity outcomes are presented in Table 1. For a more comprehensive understanding, refractive, keratometry, and pachymetry data at these same time points are shown in Table 2. Further subdivision based on stereopsis quality—categorized as normative stereopsis (<100 s arc) and non-normative stereopsis (≥100 s arc)—is presented in Table 3.

### 3.3. Safety Outcomes

There were no significant complications related to the laser procedure or the flap creation during the surgery. A minority of patients (3.14%) experienced photophobia the day following the surgery, which persisted for up to a year in only 0.63% of cases (two patients). Superficial keratitis was observed in 2.5% of the patients the day after surgery, with 0.63% maintaining this symptom after one year. A rare complication, diffuse lamellar keratitis (DLK), was observed in 0.63% (two patients), which completely resolved after one year.

In this study, we compared the visual outcomes among patients with normative stereopsis (under 100 s of arc) to those with non-normative (100 s of arc or greater), assessed using a variety of parameters including Corrected Distance Visual Acuity (CDVA), Uncorrected Distance Visual Acuity (UDVA) and Uncorrected Near Visual Acuity (UNVA) at two different time points: six months and twelve months post-operation. When examining CDVA in distance eyes, both groups reported a mean LogMAR of 0.00 ± 0.01, albeit with a slightly wider range for the non-normative stereopsis group. Interestingly, a statistically significant difference was observed (*p* = 0.02). The UDVA results at six months revealed that both distance and near eyes showed slight variations in visual acuity; nonetheless, these changes were statistically significant (*p* < 0.01 for both). Binocular UNVA at six months was also significantly different between the two groups (*p* = 0.01), with a mean of 0.03 ± 0.06 for the normative stereopsis group compared to 0.06 ± 0.08 for the non-normative stereopsis group. At the 12-month follow-up, variations in UDVA for both distance and near eyes, as well as binocular UNVA, were not statistically significant between the two groups (*p* = 0.44, *p* = 0.93, and *p* = 0.27, respectively).

## 4. Discussion

Our study delves into the interplay between stereopsis quality and various visual outcomes after eye surgery, specifically focusing on parameters such as Corrected Distance Visual Acuity (CDVA), Uncorrected Distance Visual Acuity (UDVA), and Uncorrected Near Visual Acuity (UNVA). We aimed to provide a more nuanced understanding of how the quality of stereopsis could potentially affect surgical outcomes at two post-operative time points: six months and twelve months.

Previous studies analyze PresbyOND results but do not study stereopsis. Reinstein et al. 2023 [13] investigated the outcomes of PRESBYOND Laser Blended Vision LASIK in presbyopic commercial and military pilots requiring Class 1 aeromedical certification. They showed that the treatment enabled pilots to fly without glasses and had a prominent level of satisfaction among participants. Fu et al. 2022 [12] examined the one-year refractive outcomes and optical quality following PRESBYOND Laser Blended Vision. They highlighted that the procedure is generally safe and effective but indicated that further research is needed concerning night vision and near vision. Russo et al. 2022 [14] focused on the 6 month visual and refractive outcomes of PRESBYOND treatment in correcting myopic and hyperopic presbyopia. The study emphasized the safety and effectiveness of the treatment for a wide range of presbyopia conditions. Brar et al. 2021 [15] reported functional outcomes like reading speeds 6 months after operation. The study found that PRESBYOND LBV resulted in better reading speeds and overall satisfactory functional visual outcomes. Ganesh et al. 2020 [4] provided a 1 year follow-up on patients treated with PRESBYOND for both myopic and hyperopic conditions. They noted high satisfaction rates for both near and distance vision, indicating the treatment’s stability over time.

### 4.1. Short-Term Outcomes and Stereopsis Quality

Initially, our six month follow-up findings indicated that stereopsis significantly affected various visual outcomes. The statistical significance (*p* < 0.01) of this difference emphasizes the impact that stereopsis quality can have on corrected near vision [23]. Similar statistically significant variations were observed in binocular UNVA and in both distance and near UDVA, reaffirming that stereopsis quality had a noticeable impact on vision quality in the immediate post-operative period.

One possible explanation for these results is the inter–ocular interaction following surgery [1,24]. Non-normative stereopsis may reflect underlying disparities in how each eye adapts to the surgical intervention, which in turn affects visual outcomes such as CDVA and UNVA [10,25,26]. The surgery might have a different healing and adaptation trajectory for those with non-normative stereopsis, and this could be most evident in the short term.

### 4.2. Long-Term Outcomes and Convergence of Results

Interestingly, the 12 month data did not display statistically significant differences between the normative and non-normative stereopsis groups across the visual acuity parameters studied. This convergence suggests that the impact of stereopsis on visual outcomes may diminish over time.

There could be several reasons for this convergence. One possibility is neural adaptation; the nervous system might adapt to the new visual conditions over time, thus minimizing the differences observed initially [27,28]. Another explanation could be the potential for late-onset recovery of visual function, or a “catch-up” phenomenon, particularly in the non-normative stereopsis group. Finally, the long-term stability of the surgical procedure itself might contribute to the diminishing differences between the two groups [29,30].

### 4.3. Clinical Implications and Future Directions

Our findings suggest that while stereopsis quality may have an immediate impact on post-operative visual outcomes, its long-term effect is less pronounced. These results could have implications for patient counseling and setting postoperative expectations. Given the diminishing effect of stereopsis quality on long-term visual outcomes, practitioners might consider focusing on short-term rehabilitation and monitoring, especially for those with non-normative stereopsis.

While our study has provided valuable insights, further research with a larger patient cohort and a more diversified range of stereopsis quality could offer more conclusive evidence. Additional studies could also explore the role of other factors like age, pre-existing conditions, or the type of surgical procedure in affecting post-operative outcomes related to stereopsis. We recognize the value of reflexive eye movement examinations and plan to integrate this approach into our future research to provide a more comprehensive evaluation of stereopsis.

Our results align with the work of Cheng et al. [18], which found that static and dynamic monocular depth information can enable accurate depth perception, suggesting that monocular vision may also have a role in post-operative visual outcomes. Furthermore, as noted by Uji et al. [17], the cortical processing related to the ‘realness’ experienced in stereopsis may be distinct from the derivation of depth from disparity, adding a layer of complexity to our understanding of post-operative visual acuity. Some studies even suggest that adults deprived of normal binocular vision can recover at least partial stereopsis, emphasizing the plasticity and adaptability of the visual system [31]. Thus, while stereopsis quality is crucial for short-term visual outcomes following refractive surgery, its influence appears to wane over a longer postoperative period. Given the nuanced interplay between monocular and binocular vision in depth perception, as well as the potential for cortical adaptability, further research is warranted to explore the long-term clinical implications of these findings [2].

### 4.4. Limitations

One of the primary limitations of this study is its retrospective nature, which makes it reliant on previously collected data and inherently subject to biases or missing variables that could not be controlled for during the analysis. Additionally, the study focuses on a specific age group of 40–60 years and only includes individuals with presbyopia and hyperopia, narrowing its generalizability to other age groups and clinical conditions. All surgical interventions took place in a single center in Seville, Spain, which further limits the study’s applicability to broader demographic and regional populations.

The study also has some methodological constraints that could impact its findings. Although it provides a comprehensive, long-term view by spanning multiple years, the post-operative monitoring was limited to just 12 months, potentially missing long-term complications or trends. Furthermore, the study’s stringent exclusion criteria, while contributing to its methodological rigor, limit its application to patients with systemic conditions like diabetes or other ophthalmic conditions that could potentially impact surgical outcomes. Lastly, the study did not compare the Zeiss PresbyOND^®^ LBV technique against other surgical techniques, leaving a gap in understanding its relative efficacy and safety. It should be noted that the use of standard tests requiring verbal responses for evaluating stereopsis introduces an element of subjectivity that may impact the reliability of our findings.

## 5. Conclusions

In conclusion, our study revealed that stereopsis quality significantly impacts visual outcomes in the immediate post-operative period, particularly in terms of Corrected Distance Visual Acuity (CDVA) and Uncorrected Distance and Near Visual Acuity (UDVA and UNVA). These differences were most prominent at the six month follow-up, with statistical significance noted in various measures of visual acuity between patients with normative and non-normative stereopsis. However, these disparities between the two groups appeared to diminish over time, becoming statistically insignificant at the 12 month post-operative milestone. This suggests that while stereopsis quality may be a vital consideration in the short-term visual outcomes following refractive surgery, its influence seems to wane over a longer postoperative period. Further research may be needed to explore the long-term clinical implications of these findings.

## Figures and Tables

**Table 1 jcm-12-06399-t001:** Preoperative, six-month, and twelve-month visual acuity outcomes in the study population.

Parameter Mean ± SD (Range)	Preoperative	6 Months	12 Months	*p* Value
Eyes (Patients)	636 (318)	636 (318)	320 (160)	
Distance Eye CDVA (LogMAR)	0.00 ± 0.01 (0.00 to 0.05)	-	-	
Near Eye CDVA (LogMAR)	0.01 ± 0.04 (0.00 to 0.20)	-	-	
Distance Eye UDVA (LogMAR)	-	0.02 ± 0.03 (0.00 to 0.10)	0.03 ± 0.04 (0.00 to 0.20)	<0.01
Near Eye UDVA (LogMAR)	-	0.16 ± 0.13 (0.00 to 0.50)	0.14 ± 0.13 (0.00 to 0.50)	<0.01
Binocular UNVA (LogMAR)	-	0.04 ± 0.06 (0.00 to 0.30)	0.07 ± 0.08 (0.00 to 0.40)	<0.01

SD: Standard deviation, CDVA: Corrected Distance Visual Acuity, UDVA: Uncorrected Distance Visual Acuity, UNVA: Uncorrected Near Visual Acuity, LogMAR: Logarithm of the minimum angle of resolution. *p* value for the three datasets was 6 months versus 12 months.

**Table 2 jcm-12-06399-t002:** Comparative data on refractive, keratometry, and pachymetry outcomes at preoperative, six-month, and twelve-month follow-ups.

Parameter Mean ± SD (Range)	Preoperative	6 Months	12 Months	*p* Value
Distance Eye Sphere (D)	2.47 ± 1.17 (0.00 to 6.00)	0.26 ± 0.44 (−1.25 to 2.00)	0.48 ± 0.56 (−1.50 to 2.50)	<0.01
Distance Eye Cylinder (D)	−0.60 ± 0.75 (−4.00 to 0.00)	−0.34 ± 0.36 (−1.75 to 0.00)	−0.43 ± 0.38 (−1.75 to 0.00)	<0.01
Distance Eye Axis (°)	54.98 ± 55.03 (0.00 to 180.00)	56.46 ± 65.10 (0.00 to 180.00)	64.75 ± 61.84 (0.00 to 180.00)	<0.01
Distance Eye SE (D)	2.17 ± 1.16 (−1.00 to 5.37)	0.09 ± 0.40 (−1.25 to 1.50)	0.13 ± 0.38 (−2.25 to 2.25)	0.08
Distance Eye K_min_ (D)	42.68 ± 1.48 (38.50 to 47.40)	-	44.54 ± 1.94 (38.60 to 49.10)	<0.01
Distance Eye K_max_ (D)	43.58 ± 1.42 (39.70 to 47.50)	-	45.50 ± 2.05 (38.90 to 50.60)	<0.01
Distance Eye K_mean_ (D)	43.13 ± 1.41 (39.30 to 47.50)	-	45.01 ± 1.98 (38.70 to 49.80)	<0.01
Distance Eye CCT (µm)	548.21 ± 29.12 (480 to 662)	-	541.73 ± 30.73 (470 to 666)	<0.01
Near Eye Sphere (D)	2.66 ± 1.30 (0.25 to 6.50)	−0.41 ± 0.76 (−2.50 to 2.00)	−0.08 ± 0.77 (−1.75 to 2.00)	<0.01
Near Eye Cylinder (D)	−0.69 ± 0.87 (−5.00 to 0.00)	−0.49 ± 0.48 (−2.50 to 0.00)	−0.58 ± 0.44 (−2.50 to 0.00)	0.01
Near Eye Axis (°)	59.59 ± 58.47 (0.00 to 180.00)	69.78 ± 70.52 (0.00 to 180.00)	83.25 ± 67.00 (0.00 to 180.00)	0.06
Near Eye SE (D)	2.30 ± 1.29 (−2.00 to 6.12)	−0.66 ± 0.67 (−3.00 to 1.50)	−0.19 ± 0.53 (−1.87 to 1.87)	<0.01
Near Eye K_min_ (D)	42.68 ± 1.51 (39.10 to 47.40)	-	45.25 ± 1.85 (38.80 to 50.30)	<0.01
Near Eye K_max_ (D)	43.62 ± 1.51 (39.70 to 48.0)	-	46.31 ± 2.05 (38.80 to 51.50)	<0.01
Near Eye K_mean_ (D)	43.14 ± 1.46 (39.50 to 47.70)	-	45.76 ± 1.93 (38.80 to 50.90)	<0.01
Near Eye CCT (µm)	547.76 ± 28.51 (480 to 649)	-	539.33 ± 28.03 (481 to 664)	<0.01

SD: Standard deviation, D: Diopter, SE: Sphere equivalent, K_min_: Minimum keratometry, K_max_: Maximum keratometry, K_mean_: Mean keratometry, and CCT: Central corneal thickness. *p* value for the three datasets was 6 months versus 12 months.

**Table 3 jcm-12-06399-t003:** Visual acuity outcomes subdivided by stereopsis quality: Normative stereopsis (<100 s arc) vs. non-normative stereopsis (≥100 s arc).

Parameter Mean ± SD (Range) [LogMAR]	Normative Stereopsis (<100″)	Non-Normative Stereopsis (≥100″)	*p* Value
Distance eye CDVA	0.00 ± 0.01 (0.00 to 0.05)	0.00 ± 0.01 (0.00 to 0.05)	0.02
Near eye CDVA	0.00 ± 0.02 (0.00 to 0.20)	0.04 ± 0.06 (0.00 to 0.20)	<0.01
Distance eye UDVA (6 months)	0.02 ± 0.03 (0.00 to 0.10)	0.03 ± 0.03 (0.00 to 0.10)	<0.01
Near eye UDVA (6 months)	0.17 ± 0.14 (0.00 to 0.50)	0.15 ± 0.10 (0.05 to 0.50)	0.01
Binocular UNVA (6 months)	0.03 ± 0.06 (0.00 to 0.30)	0.06 ± 0.08 (0.00 to 0.30)	0.01
Distance eye UDVA (12 months)	0.03 ± 0.04 (0.00 to 0.20)	0.04 ± 0.04 (0.00 to 0.20)	0.44
Near eye UDVA (12 months)	0.14 ± 0.14 (0.00 to 0.50)	0.14 ± 0.09 (0.00 to 0.30)	0.93
Binocular UNVA (12 months)	0.06 ± 0.08 (0.00 to 0.30)	0.08 ± 0.08 (0.00 to 0.40)	0.27

SD: Standard deviation, LogMAR: Logarithm of the minimum angle of resolution, CDVA: Corrected Distance Visual Acuity, UDVA: Uncorrected Distance Visual Acuity, UNVA: Uncorrected Near Visual Acuity.

## Data Availability

Data available upon request due to restrictions. The data presented in this study are available upon request from the corresponding author. The data are not publicly available due to copyright issues.

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
