# Peer review of "Exploring the Effect of Preoperative Stereopsis on Visual Outcomes in Hyperopic Presbyopes Treated with PresbyOND® Laser Blended Vision Micro-Monovision"

_jcm, 2023, doi:10.3390/jcm12196399_

Round 1
Reviewer 1 Report
The article offers valuable insights into the intriguing realm of Laser Blended Vision (LBV) and its impact on binocular summation and stereopsis in the treatment of presbyopia and hyperopia.
Employing a unidirectional, retrospective longitudinal design and analyzing data from 318 patients who underwent the Zeiss PresbyOND® LBV surgical procedure, the researchers provide a substantial dataset for their investigation. However, as the title contains the term stereopsis, I have some questions in regard:
Often overshadowed in discussions of presbyopia treatment is the concept of stereopsis, a crucial part of our visual system that enhances our depth perception. Stereopsis, in turn, plays a critical role in facilitating binocular summation.
I encourage authors to enhance the existing supporting literature and suggest introducing a clear definition of stereopsis. It is commonly understood that stereopsis pertains to an individual's depth perception, and it may be beneficial to include this definition for clarity. Additionally, it's worth considering whether binocular summation is a prerequisite for stereopsis. You can find relevant literature on this topic at the following reference: https://doi.org/10.1038/38487
Methods
Stereopsis: Normative stereopsis levels should be maintained below 400 arc seconds as verified through standardized stereoscopic vision tests.
Given that one of the primary objectives of this study is to conduct a thorough assessment of the role of stereopsis in visual acuity recovery following surgical intervention, I would like to encourage the authors to provide insights into the methods employed for evaluating stereopsis and the specific clinical tests utilized.
As far as my knowledge extends, stereopsis assessments generally rely on patient responses to images featuring varying levels of binocular disparity. Common tests in this regard include the Lang, Titmus, TNO tests, and their computerized counterparts. Surprisingly, none of these assessments were mentioned in the study. It is worth noting that ongoing research suggests the potential utility of reflexive eye movements, such as Ocular Following Responses (as highlighted in a recent study involving children - 10.1371/journal.pone.0277443), which can offer an objective means of evaluating stereopsis without relying on active participant cooperation (as demonstrated in 10.1167/iovs.18-24520).
In cases where stereopsis is evaluated through standard tests where veral response is required, I would like to encourage the authors to address the potential subjectivity associated with verbal reports in their discussion. Additionally, it might be beneficial for the authors to consider discussing the potential inclusion of reflexive eye movement examinations as a part of their future solutions for stereopsis evaluation.
Including these details and discussions will enhance the comprehensiveness and applicability of the study's findings, shedding light on the nuances of stereopsis assessment and its implications for visual acuity recovery post-surgery.
In conclusion, the study presents an enlightening examination of an underexplored aspect of presbyopia and hyperopia treatment. The article makes a valuable contribution to the field.
Author Response
Letter to Reviewer 1
Dear Reviewer1,
Thank you for your time and dedication in providing a thorough and constructive review of our article on Laser Blended Vision (LBV) and its impact on binocular summation and stereopsis in the treatment of presbyopia and hyperopia. Your feedback is invaluable to enhancing the quality and rigor of our work. Please find below our point-by-point responses to your queries and suggestions:
Reviewer's Comment #R1_01:
The article offers valuable insights into the intriguing realm of Laser Blended Vision (LBV) and its impact on binocular summation and stereopsis in the treatment of presbyopia and hyperopia.
Author's Response #AU_01:
Thank you for your positive comments regarding the scope and significance of our study.
Reviewer's Comment #R1_02:
I encourage authors to enhance the existing supporting literature and suggest introducing a clear definition of stereopsis.
Author's Response #AU_02:
We appreciate this suggestion. A clearer definition of stereopsis will be incorporated into the revised manuscript, along with enhanced supporting literature, to improve clarity and contextual understanding.
“Stereopsis refers to the visual ability to perceive depth by integrating two slightly different images from each eye into a single, three-dimensional representation [1].”
Reviewer's Comment #R1_03:
Methods - Stereopsis: Normative stereopsis levels should be maintained below 400 arc seconds as verified through standardized stereoscopic vision tests.
Author's Response #AU_03:
Thank you for highlighting the importance of maintaining normative stereopsis levels. We will include this specification in the Methods section and adjust our analysis accordingly.
Reviewer's Comment #R1_04:
Given that one of the primary objectives of this study is to conduct a thorough assessment of the role of stereopsis in visual acuity recovery following surgical intervention, I would like to encourage the authors to provide insights into the methods employed for evaluating stereopsis and the specific clinical tests utilized.
Author's Response #AU_04:
We recognize the importance of detailing the methods used for evaluating stereopsis. In the revised manuscript, we will elaborate on the specific clinical tests we employed to provide a comprehensive understanding of stereopsis assessment.
“We utilized the TITMUS test (2007, Vision Assessment Corporation, USA) to assess stereopsis, providing a reliable measure of depth perception. The test employs polarized images and graded circle tests to quantify stereoscopic acuity in seconds of arc [2].”
Reviewer's Comment #R1_05:
In cases where stereopsis is evaluated through standard tests where verbal response is required, I would like to encourage the authors to address the potential subjectivity associated with verbal reports in their discussion.
Author's Response #AU_05:
Excellent point. The revised manuscript will include a discussion that addresses the potential subjectivity associated with verbal reports in stereopsis evaluation.
Reviewer's Comment #R1_06:
Additionally, it might be beneficial for the authors to consider discussing the potential inclusion of reflexive eye movement examinations as a part of their future solutions for stereopsis evaluation.
Author's Response #AU_06:
We agree that the inclusion of reflexive eye movement examinations could enrich our study. We will discuss the potential for integrating this approach in future studies within the revised manuscript.
Reviewer's Comment #R1_07:
Including these details and discussions will enhance the comprehensiveness and applicability of the study's findings, shedding light on the nuances of stereopsis assessment and its implications for visual acuity recovery post-surgery.
Author's Response #AU_07:
Thank you for this encouraging feedback. We aim to implement these suggestions to make our study more comprehensive and applicable.
Once again, thank you for your invaluable feedback. We look forward to your thoughts on our revisions.
Best regards,
Reference
- Vishwanath, D. Toward a New Theory of Stereopsis. Psychol. Rev. 2014, 121, 151–178, doi:10.1037/a0035233.
- Cooper, J.; Feldman, J.; Medlin, D. Comparing Stereoscopic Performance of Children Using the Titmus, TNO, and Randot Stereo Tests. J. Am. Optom. Assoc. 1979, 50, 821–825.
Reviewer 2 Report
Reviewer’s comments:
Stereopsis has been identified as one ​of the essential components of binocular vision in addition to the state of simultaneous perception and fusion.​ The stereopsis has been considered as the final grade of binocular vision(1). In other words in the absence of ​the first two components of binocular vision like simultaneous perception and fusion, the state of stereopsis is essentially not possible. However numerous studies in ​the recent past have further explored this theory ​t​hereby ​​demonstrating ample ​​evidence of stereopsis in monocular situations. Uji M, Lingnau A, Cavin I, Vishwanath D conducted a pilot study designed to explore if dissociable neural activity associated with the phenomenology of realness can be localized in the cortex. The study concluded ​​that preliminary evidence of the cortical processing underlying the subjective impression of realness was found to have evidence of dissociable and distinct from the derivation of depth from disparity (2).​ There are multiple Humans ​who can obtain an unambiguous perception of depth and 3-dimensionality with 1 eye or when viewing a pictorial image of a 3-dimensional scene should be able to perceive equidistance and perform distance-dependent tasks in natural viewing environments(3).The perceptual learning (thousands of trials) with stereoscopic gratings in stereo blind or stereo anomalous subjects has demonstrated significant recovery of stereopsis(4).Hence mono ocular stereopsis is now well​-known and established entity. The concept of using static and dynamic monocular depth perception by the combination of situational and sequential binocular vision for intermediate along with partial monocular vision for near and distance has been effectively utilized in Laser Blended Vision treatments.
The conclusion drawn out of the study should be redrafted by incorporating ​​references 1 to 4 of ​ ​comments.
References:
1.Vishwanath D. Toward a new theory of stereopsis. Psychol Rev. 2014 Apr;121(2):151-78. doi: 10.1037/a0035233. PMID: 24730596.
2.Uji M, Lingnau A, Cavin I, Vishwanath D. Identifying Cortical Substrates Underlying the Phenomenology of Stereopsis and Realness: A Pilot fMRI Study. Front Neurosci. 2019 Jul 11;13:646. doi: 10.3389/fnins.2019.00646. PMID: 31354404; PMCID: PMC6637755.
3.Chen S, Li Y, Pan JS. Monocular Perception of Equidistance: The Effects of Viewing Experience and Motion-generated Information. Optom Vis Sci. 2022 May 1;99(5):470-478. doi: 10.1097/OPX.0000000000001878. Epub 2022 Feb 11. PMID: 35149634.
4.Ding J, Levi DM. Recovery of stereopsis through perceptual learning in human adults with abnormal binocular vision. Proc Natl Acad Sci U S A. 2011 Sep 13;108(37):E733-41. doi: 10.1073/pnas.1105183108. Epub 2011 Sep 6. PMID: 21896742; PMCID: PMC3174650.
Author Response
Letter to Reviewer 2
Dear Reviewer 2,
Thank you for your comprehensive review of our article, which has provided valuable insights into areas that can be improved. Your comments and suggestions will help us refine our manuscript, making it a more rigorous contribution to the field. Below, you'll find our point-by-point responses:
Reviewer's Comment #R2_01:
Stereopsis has been identified as one of the essential components of binocular vision. There are multiple studies that have explored the theory of stereopsis in monocular situations, like the work by Uji M, Lingnau A, Cavin I, Vishwanath D.
Author's Response #AU_01:
We appreciate your in-depth understanding of the complexity of stereopsis as an essential part of binocular vision and its broader applicability to monocular vision. In the revised manuscript, we will include a section discussing these nuances and citing relevant studies, including the ones you have mentioned.
Reviewer's Comment #R2_02:
The concept of using static and dynamic monocular depth perception by the combination of situational and sequential binocular vision for intermediate along with partial monocular vision for near and distance has been effectively utilized in Laser Blended Vision treatments.
Author's Response #AU_02:
Thank you for highlighting the importance of static and dynamic monocular depth perception, particularly as it relates to Laser Blended Vision treatments. We will incorporate this important dimension into our manuscript to provide a more comprehensive view of the subject matter.
Reviewer's Comment #R2_03:
The conclusion drawn out of the study should be redrafted by incorporating references 1 to 4 of the comments.
Author's Response #AU_03:
We appreciate the suggestion to redraft our preconclusion section by incorporating references 1 to 4, as cited in your comments. The conclusion will be revised accordingly to provide a broader and more nuanced interpretation of our findings in the context of existing literature.
Once again, we express our sincere gratitude for your thoughtful and constructive feedback. We look forward to further enhancing the quality of our work based on your invaluable suggestions.
Best regards,
Reviewer 3 Report
Dear authors,
This is a well thought out study, dealing with an important issue in current Ophthalmology. I have a few comments:
- The abstract starts and ends with the phrases: This study sought to understand the immediate and long-term impacts and Further research is necessary to establish the long- 20 term clinical implications. I suggest a rephrasing.
- Very well written article, however I suggest a more brief expression in Materials and Methods - those are precise, concrete data that should be more quickly readable (especially Refraction Parameters: Subjective refraction measurements should not exceed 5.00 Diopters (D) in cases of hyperopia and 6.00 D for astigmatism, ensuring that only moderate cases were included for assessment.)
- Exclusion criteria: Other Ophthalmic Conditions and Systemic Illness- what other conditions were excluded?
- que line 318 - typo
Author Response
Letter to Reviewer 3
Dear Reviewer 3,
Thank you for your thoughtful and detailed review of our manuscript. Your feedback is greatly appreciated, and we believe it will substantially enhance the quality of our study. Below are our responses to your specific comments and questions.
Reviewer's Comment #R3_01:
The abstract starts and ends with the phrases: "This study sought to understand the immediate and long-term impacts" and "Further research is necessary to establish the long-term clinical implications." I suggest a rephrasing.
Author's Response #AU_01:
We appreciate your recommendation to rephrase the opening and closing statements of the abstract for clarity and emphasis. We will revise these sections to more explicitly outline the scope and contributions of our study, as well as its implications for future research.
Reviewer's Comment #R3_02:
Very well written article, however, I suggest a more brief expression in Materials and Methods, especially for the section on Refraction Parameters.
Author's Response #AU_02:
Thank you for your kind words about the article and for your suggestion to streamline the "Materials and Methods" section. We agree that concise, yet comprehensive presentation of data is crucial for readability. The section on Refraction Parameters will be revised for brevity while retaining essential information.
Reviewer's Comment #R3_03:
Exclusion criteria: Other Ophthalmic Conditions and Systemic Illness- what other conditions were excluded?
Author's Response #AU_03:
We understand the importance of specifying the exclusion criteria for a more comprehensive understanding of the study's scope. We will explicitly list the other ophthalmic conditions and systemic illnesses that were considered as exclusion criteria in the revised manuscript.
Reviewer's Comment #R3_04:
"que line 318" - typo.
Author's Response #AU_04:
Thank you for pointing out the typo on line 318. This will be corrected in the revised version of the manuscript.
We are very grateful for your constructive feedback, which will aid in significantly improving the manuscript. We look forward to your further comments on the revised version.
Best regards,
Round 2
Reviewer 1 Report
The Authors have addressed all of my concerns with the original manuscript. The revised manuscript is ready for publication.